# Identification of inulin-responsive bacteria in the gut microbiota via multi-modal activity-based sorting

Alessandra Riva[1,13], Hamid Rasoulimehrabani [1,2], José Manuel Cruz-Rubio [3], Stephanie L. Schnorr[1], Cornelia von Baeckmann[4], Deniz Inan[1], Georgi Nikolov[1], Craig W. Herbold [1], Bela Hausmann [5,6], Petra Pjevac [1,5], Arno Schintlmeister[1], Andreas Spittler [7], Márton Palatinszky [1], Aida Kadunic[1], Norbert Hieger[1], Giorgia Del Favero [8], Martin von Bergen [9], Nico Jehmlich [9], Margarete Watzka[10], Kang Soo Lee [11], Julia Wiesenbauer[2,10], Sanaz Khadem[1], Helmut Viernstein[3], Roman Stocker [11], Michael Wagner [1,12], Christina Kaiser [10], Andreas Richter [10], Freddy Kleitz [4] & David Berry [1,5] ✉

Prebiotics are defined as non-digestible dietary components that promote the growth of beneficial gut microorganisms. In many cases, however, this capability is not systematically evaluated. Here, we develop a methodology for determining prebiotic-responsive bacteria using the popular dietary supplement inulin. We first identify microbes with a capacity to bind inulin using mesoporous silica nanoparticles functionalized with inulin. 16S rRNA gene amplicon sequencing of sorted cells revealed that the ability to bind inulin was widespread in the microbiota. We further evaluate which taxa are metabolically stimulated by inulin and find that diverse taxa from the phyla *Firmicutes* and *Actinobacteria* respond to inulin, and several isolates of these taxa can degrade inulin. Incubation with another prebiotic, xylooligosaccharides (XOS), in contrast, shows a more robust bifidogenic effect. Interestingly, the *Coriobacteriia Eggerthella lenta* and *Gordonibacter urolithinfaciens* are indirectly stimulated by the inulin degradation process, expanding our knowledge of inulin-responsive bacteria.

Non-digestible fibers are dietary components important for maintaining gut microbiome diversity and metabolic function[1,2]. Some dietary fibers have been shown to confer a variety of health benefits, which has led to their classification as prebiotics[2]. The original definition of prebiotics, introduced in 1995, was "non-digestible food ingredients that beneficially affect the host by selectivity stimulating the growth and/or activity of one or a limited number of bacterial species already resident in the colon, and thus attempt to improve host health"[3]. The initial concept underwent many revisions, and a consensus was achieved in 2016, defining a prebiotic as "a substrate that is selectively utilized by host microorganisms conferring a health

benefit"[4]. Although all currently recognized prebiotics are fibers, not all fibers are prebiotics. Classification of a food ingredient as a prebiotic requires scientific demonstration that the ingredient: (1) resists gastric acidity, hydrolysis by mammalian enzymes, and absorption in the upper gastrointestinal tract, (2) is fermented by the intestinal microbiota, and (3) selectively stimulates the growth and/or activity of intestinal bacteria potentially associated with health and well-being of the host[5,6]. Inulin, one of the most popular prebiotics in the food and supplement industry, is a non-structural polysaccharide produced by plants consisting of <60 linearly β−1,2-linked D-fructosyl residues with a terminal α−1,2-linked D-glucose moiety. Inulin is

abundant in foods such as banana, chicory root, Jerusalem artichoke, wheat, barley, rye, onions, leeks, and garlic[7]. Inulin consumption from natural sources is estimated to be approximately 2–11 g/d in most European countries and 2–8 g/d in the United States[8,9]. Positive effects on human health ascribed to inulin include reduction in insulin resistance[10,11], anti-inflammatory activity[11,12], and anti-carcinogenic properties[13].

Inulin is not digestible by humans, and after ingestion, it enters the large intestine where it is degraded and fermented by the gut microbiota[6]. Microbial inulinases catalyze the hydrolysis of inulin, producing fructooligosaccharides (FOS) and glucose and fructose monomers. Most characterized inulinases are of fungal origin and are used for industrial food production[14]. Information on gut bacterial inulinases is scarce, and thus far only members of the genera *Bacteroides*, *Lactobacillus* and *Bifidobacterium*, and *Arabiibacter massiliensis* have experimentally validated inulinase activity[15–18]. An increase in the relative abundance of *Bifidobacterium* upon inulin consumption has been reported in human intervention studies[19–22]. Although dietary intervention with prebiotics is a promising strategy for modulating the gut microbiota, gaps in knowledge regarding the microbial metabolic pathways involved in the utilization of prebiotics[23], inter-individual variability[19,24,25] and strain-level diversity has made the design of rational interventions challenging[23].

Due to its health benefits as well as its popularity, a better understanding of inulin metabolism by the gut microbiota is of great importance. At the same time, in-depth knowledge about the interaction between gut microbiota and inulin would enhance our understanding of prebiotic function and the development of science-based dietary interventions. To assess which bacterial taxa interact with inulin or are stimulated by it or its breakdown products, we employed ex vivo gut microbiota incubations and a multi-modal sorting approach that relies on measurement of fluorescence and Raman signals followed by 16S rRNA gene amplicon sequencing and targeted isolation of microbial strains. We validated our findings with physiological experiments, fluorescence in situ hybridization (FISH), and whole genome sequencing and analysis (Supplementary Fig. 1). This revealed that a diverse set of bacteria were able to interact with or use inulin, including previously unrecognized species, thereby demonstrating its properties as a broadly stimulating microbial nutritional substrate.

## Results

### Detection of inulin-binding bacteria in the gut microbiota
Many polysaccharides are too large to be directly imported into the cell, so they must either first be partially hydrolyzed by secreted enzymes or captured by cell surface proteins with carbohydrate-binding modules[26–29], although the import of fructans without surface pre-digestion by *Bacteroides* spp. has also been described[23].

In order to identify members of the gut microbiota with inulin-binding characteristics, we employed fluorescently labeled mesoporous silica nanoparticles (MSNs) grafted with inulin[30,31] to capture and sort inulin-bound bacteria. We performed anaerobic incubations of freshly collected human stool samples from six different donors and incubated them for 1 h in the presence of inulin-grafted or ungrafted MSNs (Fig. 1a). One hour was chosen to allow sufficient time for attachment but to minimize the possibility of complete degradation of inulin. Two different formulations of MSNs—one hydrophilic and the other hydrophobic—were used to maximize the capture of bacteria with different cell surface properties. Structured illumination confocal microscopy (SIM) confirmed that inulin-grafted MSNs possessed a high affinity for bacteria (Fig. 1a). Specific binding of bacteria to inulin-grafted MSNs was quantified by flow cytometry, which showed an increase in the number of DAPI-stained bacteria bound to inulin-grafted MSNs compared to ungrafted controls (hydrophilic MSNs: mean increase ± sd, 70 ± 4.4%, Student's *t*-test: $p < 0.0001$,

hydrophobic MSNs: mean increase ± sd, 24 ± 2.5%, Student's *t*-test: $p = 0.0003$; Supplementary Fig. 2).

We next performed fluorescence-activated cell sorting (FACS) of inulin-grafted and ungrafted MSNs followed by 16S rRNA gene amplicon sequencing to determine which bacterial taxa were bound to MSNs. Stool samples had a microbial composition typical for the healthy adult gut[32] (Fig. 1b). Microbiome profiles of sorted MSNs had high technical reproducibility, with technical replicates only contributing to 1.1% of the variation in the dataset (PerMANOVA, $p = 0.576$). The largest source of variability in the dataset was due to host inter-individual differences, which accounted for 69.1% of the total variation (PerMANOVA, $p = 0.001$). This was also the main driver of the clustering of unsorted and sorted samples in principal coordinate ordination (Bray-Curtis distances, ANOSIM: $p = 0.001$; Supplementary Figs. 3 and 4a–d). Despite this inter-individual host variability, jackknife-based diversity estimates indicated that we had identified ~80% of amplicon sequence variants (ASVs) and 99% of all genera that would be found if stool from additional donors were to be analyzed (Supplementary Fig. 4e, f). This suggests that the inulin-binding fraction of the microbiota is, like the entire microbiota, individualized to a certain degree. However, much of this variation appears to be due to differences in frequency, as we also find that 40% of identified genera were detected in all six samples and 77% were present in at least half the samples (Supplementary Fig. 4g). Bacterial genera significantly enriched in inulin-grafted MSNs compared to ungrafted MSNs belonged to the families *Lachnospiraceae* (*Blautia*, *Roseburia*, and *Lachnospira*), *Ruminococcaceae* (*Ruminococcus*, *Agathobaculum*, and *Dysosmobacter*), *Lactobacillaceae* (*Lactobacillus*), *Veillonellaceae* (*Dialister*), *Selenomonadaceae* (*Mitsuokella*), *Bacteroidaceae* (*Phocaeicola*), and *Eggerthellaceae* (*Raoultibacter*) (Wald test, $p < 0.05$, $n = 156$) (Fig. 1b; Supplementary Data 1).

### Translationally active microbiota after inulin amendment
As not all bacteria that benefit from inulin might bind to it—either because they secrete extracellular inulinases or because they cross-feed on liberated sugars or metabolic products of inulin degraders, we next sought to identify gut microbes that display increased translational activity in the presence of inulin. We performed 6 h anaerobic incubations[33] of freshly collected stool samples from eleven different donors supplemented with 2 mg/ml inulin, a level similar to reasonable dietary intake[5,34], as well as the cellular activity marker L-Azidohomoalanine (AHA). The 6 h time point was chosen based on an optimization of incubation time in a previous study[33]. The presence of translationally active cells was detected following incubation using bioorthogonal non-canonical amino acid tagging (BONCAT)[35], and fluorescently labeled cells were sorted by FACS[36]. For analysis, we divided the samples into total community (samples submitted to anaerobic incubation only without FACS sorting) and active fraction (samples submitted to anaerobic incubation and BONCAT-labeled cells sorted by FACS) (Fig. 2a). The taxonomic composition of the stool microbiota was typical for that of healthy adults[32] (Fig. 2b). We confirmed that the technical variability associated with the BONCAT procedure was much smaller than the observed biological variability [(technical variation: 0.2%, PerMANOVA, $p = 0.83$; biological variation: 57%, PerMANOVA: $p = 0.001$)], indicating the reproducibility and accuracy of the method.

Translationally active cells were detected in all inulin-amended samples after incubation with inulin, while there was only minimal activity observed in incubations without inulin amendment (Supplementary Fig. 5). The diversity and the composition of the total microbial community did not change significantly during the incubations (PerMANOVA, $p = 0.591$, Supplementary Fig. 6), indicating that these experimental conditions did not appreciably modify the composition of the microbiota. However, inulin degradation was detected in all the stool samples (mean ± sd: 31.7 ± 13%)

**a)**

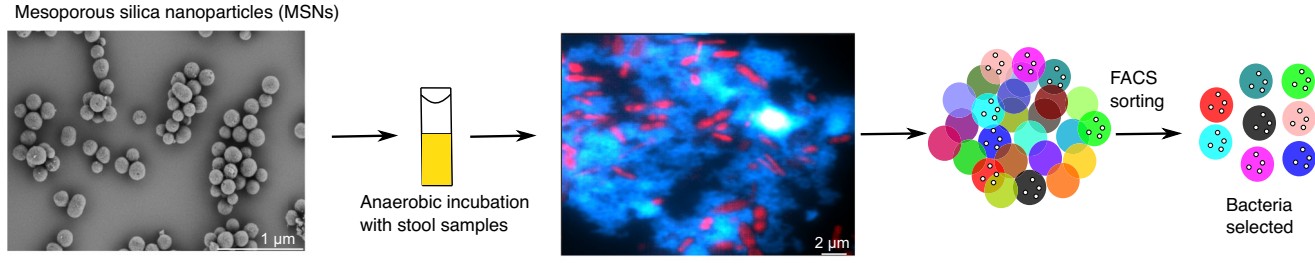

**b)**

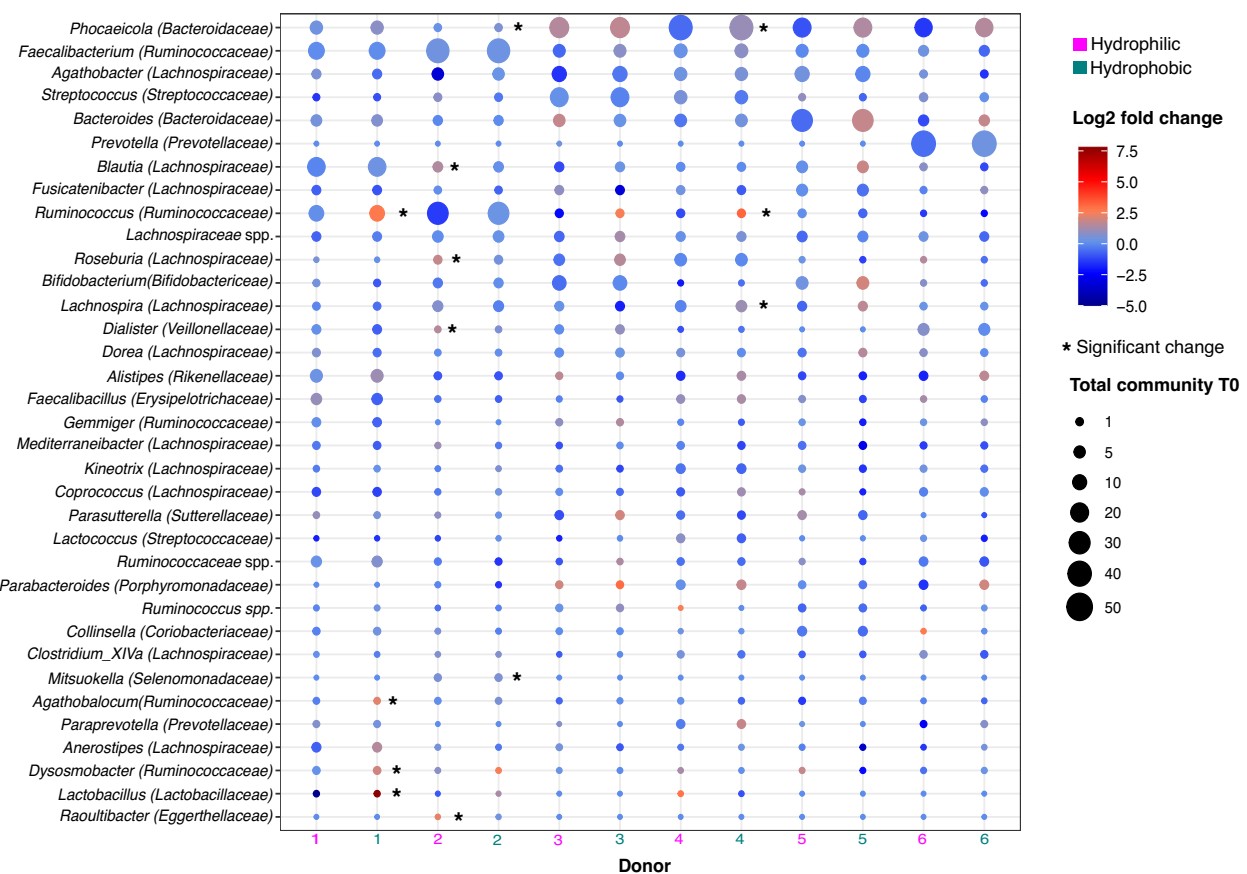

**Fig. 1 | Inulin-binding taxa in the gut microbiota. a** Experimental design: After 1 h of anaerobic incubation of human stool samples with mesoporous silica nanoparticles (MSNs, scanning electron microscopy micrograph of MSNs is shown on the left), structured illumination microscopy fluorescence image revealed the interaction between bacteria (DAPI = red) and MSNs (rhodamine = blue). Subsequently, the bacteria bound to MSNs were sorted with fluorescence-activated cell sorting (FACS) and profiled with 16S rRNA gene amplicon sequencing. Colors refer to different bacterial taxa. **b** Log2 fold changes for each donor are shown at the genus level. Bubble size indicates relative abundance in the starting sample. Asterisks indicate significantly enriched taxa after supplementation with inulin-grafted MSN, as calculated with the Wald test (2-sided, $p < 0.05$, $n = 108$ samples). Genera with relative abundance >0.5% were considered. Hydrophobic and hydrophilic refers to the type of MSNs used. Source data are provided as a Source Data file.

(Supplementary Fig. 5c) and a substantial fraction of cells were BONCAT-labeled (mean ± sd: 51.3 ± 24.3%) (Supplementary Fig. 5d). Donor 12 was the only participant with a low percentage of labeled cells, which could be due to a less abundant but extremely active inulin-degrading population. The most abundant genera in the BONCAT-positive fraction belonged to the families *Bacteroidaceae* (*Phocaeicola* and *Bacteroides*), *Lachnospiraceae* (*Blautia* and *Eubacterium*), *Ruminococcaceae* (*Faecalibacterium* and *Ruminococcus*), *Rikenellaceae* (*Alistipes*), and *Prevotellaceae* (*Prevotella*). Taxa that were enriched in the BONCAT-positive fraction compared to the total community for each participant belonged to the *Erysipelotrichaceae* (*Faecalibacillus* and *Clostridium*_XVIII),

*Coriobacteriaceae* (*Collinsella*), *Corynebacteriaceae* (*Corynebacterium*), *Lachnospiraceae* (*Blautia*, *Anaerostipes*, *Anaerobutyricum*, *Roseburia*, *Agathobacter*, and *Enterocloster*), *Bifidobacteriaceae* (*Bifidobacterium*), *Sutterellaceae* (*Parasutterella*), *Ruminococcaceae* (*Ruminococcus*), *Streptococcaceae* (*Streptococcus*) (Wald test, $p < 0.05$, $n = 33$). The most consistent enrichments across donors were found for *Blautia*, *Collinsella* and *Faecalibacillus* (Fig. 2b; Supplementary Data 2). Similarly to the MSN experiments, there was substantial inter-individual variability in the BONCAT-enriched fraction due to variations in the frequency of different taxa and, to a lesser extent to variations in taxon presence/absence, though jackknife-based diversity estimates (Supplementary Fig. 7a–d)

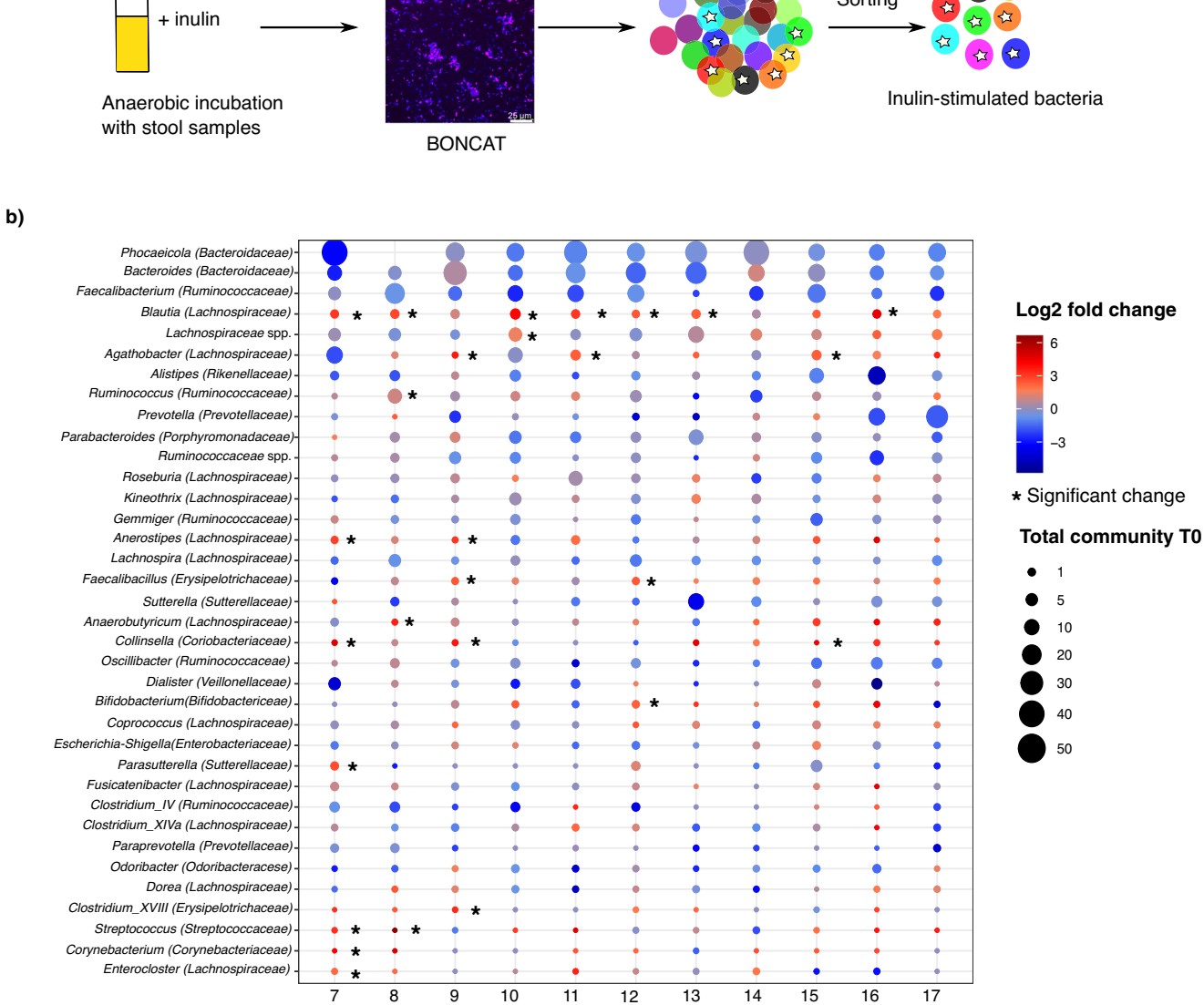

**Fig. 2 | Translationally active community after inulin supplementation. a** Eleven fresh stool samples were incubated anaerobically in the presence of the prebiotic inulin. The 11 donors were different from the 6 donors enrolled for the MSN experiment. After 6 h, translationally active cells were sorted with BONCAT-FACS based on Cy5 fluorescence signals (pink). Color code refers to different bacteria taxa and stars represent translationally active fraction sorted. **b** Log2 fold changes for each donor are shown at the genus level. Bubble size indicates relative abundance in the starting sample. Asterisks indicate significantly enriched taxa after supplementation with inulin, as calculated with the Wald test (2-sided, $p < 0.05$, $n = 33$ samples). Genera with relative abundance >0.5% were considered. Source data are provided as a Source Data file.

indicated that we had identified ~76% of ASVs and 99% of all genera that would be found if stool from additional donors were to be analyzed (Supplementary Fig. 7e, f), and 30% of identified genera were detected in all eleven samples and 86% were present in at least half the samples (Supplementary Fig. 7g). In addition to evaluating the response to native inulin with a degree of polymerization (DP) of more than 10, the same experimental approach was applied to the inulin breakdown products fructooligosaccharides (FOS, <10 monomeric units) and fructose. Comparing the active fraction and the total community after inulin, FOS, and fructose supplementations, inulin activated the largest number of ASVs, and this activation pattern had only partial overlap with the one observed after the addition of smaller compounds (Supplementary Fig. 8, Supplementary Data 2–4, Supplementary Table 1), suggesting that DP is an important factor in the gut microbiota response to fructans.

## Targeted isolation of inulin-stimulated bacteria using Raman-activated cell sorting

As a large diversity of microorganisms were found to be stimulated by inulin, we next evaluated their ability to degrade inulin using targeted isolation and pure culture physiological experiments (Fig. 3a). For this purpose, 50% heavy water ($D_2O$)-containing medium was used for the incubations as a universal marker for cellular activity, as active cells incorporate deuterium (D) from heavy water into their biomass, resulting in carbon-deuterium (C-D) bonds that can be detected by Raman microspectroscopy[37]. Inulin supplementation of fecal samples resulted in metabolically active cells incorporating D from $D_2O$ above an established threshold level (measured as %CD) (Fig. 3b). Control samples incubated with $D_2O$ but not supplemented with inulin showed only negligible levels of D incorporation. Between 47 and 100% of cells from incubations were identified as D-labeled (Fig. 3c), highlighting the

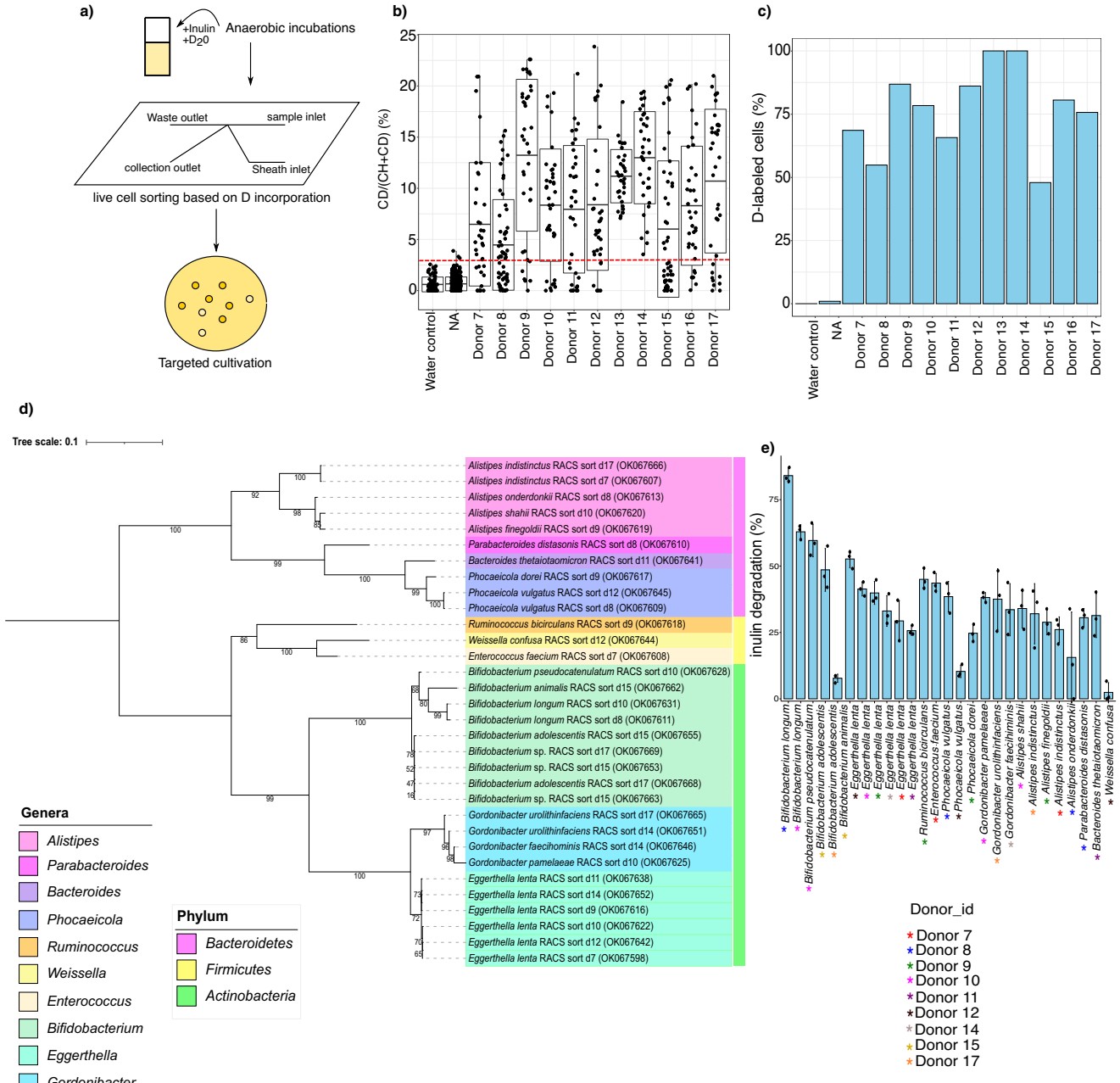

**Fig. 3 | RACS isolated strains are involved in inulin degradation. a** Experimental design. **b** Percentage of active cells measured by Raman microspectroscopy. The *x*-axis represents stool samples collected from 11 donors and incubated for 6 h in D₂O-containing media in the presence of inulin. The *y*-axis shows the level of D incorporation of randomly selected cells, as quantified by %CD. Each dot represents a random single-cell measurement (water control: $n = 76$ cells, nothing added NA: $n = 313$ cells, donor 7: $n = 35$ cells, donor 8: $n = 62$ cells, donor 9: $n = 38$ cells, donor 10: $n = 37$ cells, donor 11: $n = 35$ cells, donor 12: $n = 36$ cells, donor 13: $n = 33$ cells, donor 14: $n = 35$ cells, donor 15: $n = 48$ cells, donor 16: $n = 36$ cells, donor 17: $n = 37$ cells). The red dashed line at 2.75% indicates the threshold for considering a cell labeled. It was determined by calculating the mean +3 sd of %CD in randomly selected cells from a stool sample incubated without the addition of D₂O (water control). Boxplot: boxplot medians (center lines), interquartile ranges (box ranges), whisker ranges. **c** Percentage of cells labeled (i.e., with %CD higher than threshold) per donor. **d** Phylogenetic analysis of representative strains isolated with RACS (OK067598-OK067669). Representative strains were selected as unique species isolated from each donor. The tree was generated with the maximum likelihood algorithm using IQtree, with the optimal model identified by modelFinder as K2P+I +G4 and rooted at mid-point. Each color in the tree denotes the different genera isolated and each color code represented as a color strip shows the different phyla (*Bacteroidetes*: pink, *Firmicutes*: yellow, *Actinobacteria*: green). **e** Inulin degradation of RACS isolates measured as the percentage of inulin degraded by each strain after incubation in inulin-supplemented media (YCFA-IN). Strains were sampled in the early stationary phase. Donors are underlined as colored asterisks in the figure. Triplicates measurements are shown. Error bars represent the standard deviation of the mean. Source data are provided as a Source Data file.

ability of inulin to broadly stimulate the microbiota. We then sorted D-labeled cells from the incubations using Raman-activated cell sorting (RACS)[38] (Supplementary Table 2, Supplementary Fig. 9) and isolated pure cultures by plating the sorted fraction on YCFA-based plates to recover the greatest diversity of microbes possible[39]

(Supplementary Table 3). In total we isolated 72 colonies from 9 of the 11 donors (4–19 colonies per sample; Supplementary Table 2) consisting of 10 bacterial genera and 18 species belonging to the phyla *Bacteroidetes* (*Bacteroides*, *Phocaeicola*, *Parabacteroide*, *Alistipes*), *Firmicutes* (*Ruminococcus*, *Weissella*, *Enterococcus*), and *Actinobacteria*

(*Eggerthella*, *Gordonibacter*, *Bifidobacterium*) (Fig. 3d, Supplementary Table 3, Supplementary Data 5). *Bifidobacterium* spp. and *Eggerthella lenta* were the dominant isolates, with *Eggerthella lenta* being isolated from six different individuals (Fig. 3d, Supplementary Table 3, Supplementary Data 5). A phylogenetic tree of the 28 representative isolates is shown in Fig. 3d. Most of the 28 representative strains were able to degrade inulin (Fig. 3e) and inulin-supplemented media boosted the growth of almost all strains (Supplementary Table 3). Inulin degradation also led to the production of low DP oligosaccharides and monosaccharides (Supplementary Fig. 10, Supplementary Table 3), which could potentially be used as a microbial "public good" by cross-feeding microbes in a complex community[40]. *Coriobacteriia* were among the most frequently recovered taxa from the RACS sorting and represented a significant proportion of D-labeled cells using Raman microspectroscopy combined with fluorescence in situ hybridization (FISH; *Collinsella* and *Coriobacterium*: 62.2 ± 27.0% and *E. lenta*: 41.0 ± 17.0% of all D-labeled cells; Supplementary Figs. 11 and 12). *E. lenta* and *Gordonibacter urolithinfaciens* isolates were also closely related to ASV sequences recovered from BONCAT-FACS (Supplementary Fig. 13, Supplementary Data 6). Although genome analysis revealed the presence of putative glycoside hydrolases (Supplementary Figs. 14 and 15, Supplementary Table 4), no enzymes with high homology to characterized inulinases were detected. Additionally, incubation of these strains with inulin did not increase their growth rate (Supplementary Fig. 16), indicating that inulin and its component sugars are not utilized to an appreciable extent for growth.

### Translationally active microbiota and targeted isolation of XOS-stimulated bacteria

To compare the stimulation of the microbiota by inulin with another candidate prebiotic, we performed the same experimental approach using the prebiotic xylooligosaccharides (XOS). We incubated an additional 6 samples in the presence of XOS, AHA and heavy water and sorted the XOS-active fractions by FACS and RACS. As reported for inulin and MSN-grafted inulin experiments, microbiome profiles had high technical reproducibility, with technical replicates only contributing to 1% of the variation in the dataset (PerMANOVA, $p = 0.671$). The largest source of variability in the dataset was due to host inter-individual differences, which accounted for 51% of total variation (PerMANOVA, $p = 0.001$, ANOSIM: $p = 0.001$ and Bray-Curtis distances Supplementary Fig. 17a–d). Jackknife-based diversity estimates indicated that we had identified 80% of amplicon sequence variants (ASVs) and 93% of all genera that would be found if stool from additional donors were to be analyzed (Supplementary Fig. 17e, f). This suggests that the XOS-active fraction of the microbiota is individualized similarly as in the inulin experiments. Part of this variation appears to be due to differences in frequency, as we also find that 35% of identified genera were detected in all six samples and 68% were present in at least half the samples (Supplementary Fig. 17g).

After 6 h incubation, XOS consistently stimulated the metabolic activity of *Bifidobacterium*, indicating a strong bifidogenic effect of XOS, as well as low-abundant *Alcaligenes* (Fig. 4a–c; Supplementary Data 7). In agreement with this observation, the use of heavy water combined with RACS allowed us to directly isolate members of the genera *Bifidobacterium* spp. (45 isolates) and *Collinsella aerofaciens* (4 isolated) as XOS utilizers (Fig. 4d, e; Supplementary Table 5, Supplementary Data 8). Unlike for inulin, XOS-supplemented media boosted the growth of all the isolates (Supplementary Fig. 18, Supplementary Table 6) and high-performance anion-exchange chromatography (HPAEC) showed a significant decreased level of xylobiose and xylotriose of the stool sample measured at 0 and 6 h (Student's $t$-test; D-xylose $p = 0.475$, xylobiose $p < 0.0001$, xylotriose $p < 0.0001$) (Supplementary Fig. 19a) as well as a significant decrease of xylobiose and xylotriose for all the strains isolated with RACS (Student's $t$-test or Wilcox test; D-xylose $p = 0.876$, xylobiose $p < 0.0001$, xylotriose

$p < 0.0001$) (Supplementary Fig. 19b) indicating XOS degradation after 6 h incubation of the stool sample and after 24 h incubation with XOS-supplemented media for all the isolates.

## Discussion

Polysaccharides can be either hydrolyzed by secreted enzymes or captured by cell surface proteins with carbohydrate-binding modules prior to import[26–29]. Prebiotics are food constituents or dietary supplements reported to promote the growth of certain gut bacteria (mainly *Bifidobacterium*, and secondarily *Lactobacillus*, *Anaerostipes*, and *Faecalibacterium prausnitzii*)[41]. Generally, however, this capability is not systematically evaluated and molecular pathways sustaining the health-claims remain largely unexplored. In this study, we investigated the capability of the microbiota response to the widely used dietary supplement inulin combining multi-modal cell sorting methods (MSN-FACS, BONCAT-FACS, and RACS), physiological experiments, stable isotope probing (SIP), FISH, and genomic analyses.

Using inulin-grafted nanoparticles, we generated highly selective nanoprobes which enabled us to discover that many taxa are capable of binding inulin. We also found widespread translational activation of the microbiota due to inulin, with the most prominent inulin-responsive taxa being members of *Lachnospiraceae* including *Blautia*, as well as *Collinsella* (*Coriobacteriaceae*) and *Faecalibacillus* (*Erysipelotrichaceae*). BONCAT-FACS and MSN-FACS are powerful methods to target both inulin-binding and active fractions of the gut microbiota. There was some concordance between inulin-binding and translational activation assays, most notably for *Blautia*, *Ruminococcus*, and *Roseburia*, suggesting that inulin binding prior to hydrolysis is a common strategy employed by gut bacteria. These observations are in accordance with previous work where strain-specific binding to glycans using glass beads was described[42]. However, the physiological effects and health benefits resulting from prebiotics utilization may be explained not only by changes in gut microbes abundance but also by changes in microbiota functionality or metabolism, which need to be taken into account for future studies[41].

Inulin response was individualized in both the translationally active and the inulin-binding microbiota fraction. The existence of a substantial inter-individual variability in gut microbiome in response to dietary interventions is becoming increasingly evident[19,24]. In a previous study both the effect of resistant starch and its magnitude varied among individuals and indicated the need for personalized strategies in the modulation of gut microbiota[25]. Another human study highlighted a large inter-individual variation in the architecture of the gut microbiome in response to inulin supplementation[24] and in an animal study inulin did not alter overall microbiota composition but was able to induce donor-specific changes in the microbiota composition[43].

To gain insights into the physiology of inulin-responsive microbes, we performed targeted isolations with RACS (based on the deuterium signal) from stool samples from 11 donors. We recovered 72 colonies belonging to 18 different species, including *Eggerthella lenta*, *Bifidobacterium* spp. *Gordonibacter* spp., *Alistipes* spp., and *Phocaeicola*. Notably, *Bifidobacterium* spp. and *Ruminococcus* were also detected with BONCAT-MSN-FACS, *Eggerthellaceae* and *Bacteroidaceae* were enriched using both MSN-FACS and RACS. Dietary intervention studies, either performed in humans, laboratory animals, or in vitro, often report an increase in *Bifidobacterium* spp[19]. upon inulin supplementation, although increases in other taxa, such as *Anaerostipes*, *Lactobacillus*, *Parabacteroides*, *Blautia*, and *Collinsella* have also been reported[20,41,43–47]. Many of these studies include non-healthy participants and additionally are unable to distinguish the effects of inulin on host gastrointestinal function from direct effects on the microbiome. Another factor to take into account is that fructan size (DP) can impact incubation time and therefore site of

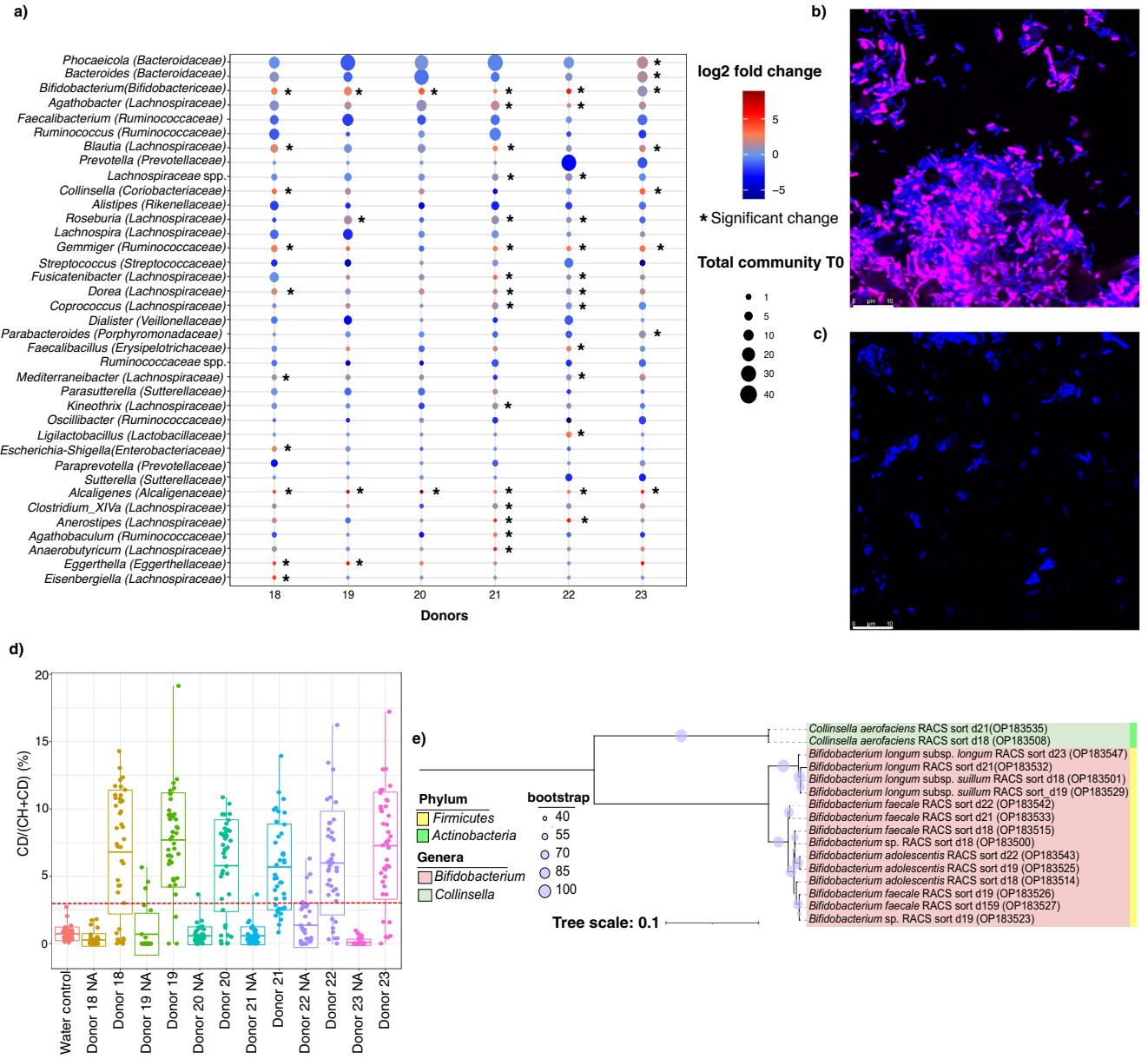

**Fig. 4 | Translationally active community and RACS isolated strains after XOS supplementation. a** BONCAT-FACS: log2 fold changes for each donor are shown at the genus level. Bubble size indicates relative abundance in the starting sample. Asterisks indicate significantly enriched taxa after supplementation with XOS, as calculated with the Wald test (2-sided, $p < 0.05$, $n = 36$ samples). Genera with relative abundance >0.5% were considered. **b** Example of fluorescence microscopy images showing BONCAT-positive cells (pink, Cy5) after 6 h incubation with XOS. All cells are stained with DAPI (blue). **c** No amendment control shows no BONCAT signal. Both images were recorded using identical settings, and imaging was performed for all 6 donors. **d** Percentage of active cells measured by Raman microspectroscopy. The x-axis represents 6 stool samples collected from 6 donors and incubated for 6 h in $D_2O$-containing media in the presence of XOS. The y-axis represents the level of D incorporation of randomly selected cells measured by %

CD. Each dot represents a random single-cell measurement (water control: $n = 40$ cells, donor 18 nothing added (NA): $n = 40$ cells, donor 18: $n = 40$ cells, donor 19 NA: $n = 30$ cells, donor 19: $n = 40$ cells, donor 20 NA: $n = 40$ cells, donor 20: $n = 40$ cells, donor 21 NA: $n = 40$ cells, donor 21: $n = 40$ cells, donor 22 NA: $n = 40$ cells, donor 22: $n = 40$ cells, donor 23 NA: $n = 40$ cells, donor 23: $n = 40$ cells). The red dashed line at 2.55% indicates the threshold for considering a cell labeled. Boxplot: boxplot medians (center lines), interquartile ranges (box ranges), whisker ranges. **e** Phylogenetic analysis of representative strains isolated with RACS after XOS supplementation (OP183499-OP183547). The tree was generated with the maximum likelihood algorithm using IQtree, with the optimal model identified by modelFinder as TN+F+G4 and rooted at mid-point. Each color in the tree denotes the different genera isolated and each color code represented as a color strip shows the different phyla. Source data are provided as a Source Data file.

fermentation in the gut, with FOS being more quickly fermented than inulin[48]. Long-chain fructans such as inulin may be fermented over a larger portion of the colon[48] where the microbial community may slightly differ in composition[49,50]. It is important to acknowledge that our ex vivo model reflects the colon community and that different species in the small intestine may be the predominant inulin utilizers.

Our data suggest that a broad group of bacteria respond to inulin. Of note, most of the 18 isolated inulin-responsive species were able

to degrade inulin in pure culture, suggesting that although monosaccharides are liberated during the hydrolysis of inulin, there may be limited cross-feeding of these sugars by non-inulin-utilizing species. The finding that members of the *Coriobacteriia* (*Eggerthella lenta* and *Gordonibacter* spp.) were among the most frequently detected bacteria in all of our inulin assays was particularly surprising, as they are more commonly associated with polyphenol and bile acid transformations and not with polysaccharides utilization[51]. Amongst the *Coriobacteriia*,

the genera *Coriobacterium, Collinsella, Atopobium, Olsenella, Enorma, Arabiibacter massiliensis* are glucose-fermenting bacteria[18,52,53], whereas *Eggerthella, Paraeggerthella, Adlercreutzia, Enterorhabdus, Asaccharobacter, Denitrobacterium, Cryptobacterium, Slackia,* and *Gordonibacter* have been considered to be asaccharolytic[52], although conflicting results with respect to sugar fermentation have been reported for *Eggerthella lenta*[52–54]. It has recently been shown that a key carbon source for *E. lenta* is acetate derived from arginine catabolism[55]. We found that neither inulin, glucose, nor fructose media supplemented with arginine boosted the growth of *E.lenta* or *G. urolithinfaciens* in pure culture. It is therefore likely that these species were active in the original stool incubations due in part to metabolic interactions with other species and that they benefit indirectly from the inulin degradation process.

To complement our data, we compared our findings with inulin to another prebiotic, XOS, which is a polymer of xylose. Targeted isolations with RACS (based on the deuterium signal) identified *Bifidobacterium* spp. and *Collinsella aerofaciens* as XOS degraders. *Bifidobacterium* was enriched in all donors in both RACS and FACS, indicating the strong bifidogenic effect of XOS in accordance with other human and animal studies[56–58].

In summary, using multi-modal activity-based sorting combined with complementary techniques, we were able to identify a widespread inulin and XOS-responsive gut microbial community, including novel inulin-responsive species. This powerful approach for detecting microbial guilds in the gut microbiota can be applied to a wide variety of research questions and is a framework that should facilitate the study of many activities of interest in the microbiome and future research on science-based dietary interventions.

## Methods

### Samples collection

Fresh fecal samples were collected from 23 healthy subjects (13 females and 10 males, age mean ± sd: 29.3 ± 5.6; BMI mean ± sd: 23.2 ± 3.0). Donors who consumed antibiotics or food supplements containing probiotic/prebiotic in the previous 6 months as well as individuals suffering from chronic or acute intestinal disease were excluded from the study. All samples were self-collected using a fecal collection tube (Sarstedt, Germany). The study was approved by, and conducted in accordance, with the University of Vienna ethics committee (reference number 00161) and written informed consent was signed by all enrolled participants.

### Synthesis of silica nanoparticles

Mobile composition of matter number forty-eight (MCM-48)-type mesoporous silica nanoparticles with a diameter of 150 nm were synthesized according to previous work[30,31]. After the synthesis, the material was calcinated at 550 °C for 5 h to remove the template. The silica nanoparticles were functionalized using a post-grafting procedure. The material was kept at 150 °C under vacuum to remove adsorbed water. After that, the particles were suspended in toluene under argon at 110 °C. After stirring for 4 h, 4 mmol g$^{-1}$ silica (3-aminopropyl)triethoxysilane (APTS) (Sigma-Aldrich, Austria) was added and left stirring overnight. Finally, the material was collected by centrifugation (after reaching room temperature) and washed with toluene followed by three times washing with ethanol and dried at room temperature. Amino-functionalized MCM-48-type MSNs (0.11 g) were suspended in 15 ml dimethylformamide (DMF) (Alfa Aesar, Austria) in an ultrasonic bath. 0.0756 g of hydroxybenzotriazole (HOBt) (Sigma-Aldrich, Austria) and dissolved in 1 ml DMF, followed by the addition of 0.421 g of 2-(1H-benzotriazole-1-yl)−1,1,3,3-tetramethyluronium hexafluorophosphate (HBTU) (ACROS-Fisher, Austria). The obtained solution was added to COOH-Inulin (0.2 g, 1 eq) and the vial was washed with 3 ml DMF. The mixture was stirred at room temperature for 10 min followed by the addition of 0.193 ml

diisopropylethylamine (DIPEA) (Iris-Biotech; Germany). The particles were centrifuged for 20 min (7000×g) and mixed with the inulin solution, followed by stirring overnight. After that, the solution was centrifuged, washed three times with DMF and three times with dichloromethane (DCM) (Sigma-Aldrich, Austria) and once with ethanol to remove unreacted substances, followed by drying in air until complete dryness. To anchor rhodamine B onto silica nanoparticles, 1 mg of Rhodamine B isothiocyanate (0.00186 mmol) (Sigma-Aldrich, Austria) was suspended in 4 ml toluene. At the same time, 100 mg of the particles were stirred in 8 ml of toluene. The same equivalent of APTS (0.436 μl) was added to the rhodamine B solution resulting in a clear pink solution. Afterward, 1 ml of the obtained solution was added to the particle suspension and the reaction mixture was stirred overnight at room temperature. To collect the particles, the mixture was centrifuged (12,000×g, 20 min) and washed with ethanol until the supernatant was of clear color (approx. 5–10 times). After that, the material was left at room temperature for drying overnight.

### Ex vivo anaerobic incubations with mesoporous silica nanoparticles (MSNs)

Six fresh stool samples were immediately introduced into an anaerobic tent (85% $N_2$, 10% $CO_2$, 5% $H_2$). MSNs were freshly prepared for each experiment, resuspended in 1 ml phosphate-buffered saline (PBS) and sonicated for at least 20 min until dissolution. Afterward, the suspended nanoparticles were introduced into the anaerobic tent and were left for 30 min to allow oxygen reduction. Every reagent was introduced in the anaerobic tent the day before the experiment to ensure that they were anaerobic by the start of the experiment. Then, 10 ml PBS was added to 1 g of fecal sample. The samples were homogenized by vortexing and afterward, the homogenate was filtered using a 40 μm size filter (Corning, Germany) to remove large particles. Samples were further diluted 1:10 in PBS to avoid background noise and autofluorescence in the rhodamine signal. Fecal samples were incubated in autoclaved Hungate tubes in the presence of MSNs at 37 °C for 1 h under anaerobic conditions. MSNs without inulin were used as negative controls for each experiment. Stool samples were diluted in total 1:200 and a final concentration of 2 mg/ml of MSNs was used for FACS experiments with a 1:1 ratio between sample and MSNs. For confocal microscopy visualization, 0.2 mg/ml of MSNs were used with a 1:20 ratio between MSNs and sample. This proportion was optimal to display a homogenous distribution between bacteria and MSNs.

Part of the sample biomass was collected at 0 and 1 h incubation times and stored at −20 °C for additional DNA extractions. After the incubation time, samples were counterstained with 4′, 6-diamidino-2-phenylindole (DAPI) (Sigma-Aldrich, Austria) and used immediately for confocal microscopy visualization and FACS.

### Confocal microscopy imaging

Bacterial cells and fluorescently labeled inulin-mesoporous silica nanoparticles were visualized with a Confocal LSM Zeiss 710 equipped with ELYRA PS. 1 system for super-resolution. MSNs were chemically functionalized with rhodamine (depicted in light blue) and bacteria counterstained with DAPI (depicted in red) (Fig. 1a). Structured illumination images (SIM) were acquired with a Plan Apochromat 100X/oil objective and grid 5 rotation. For spatial distribution, at least 40 images along the Z axes were imaged in randomly chosen optical fields and every experimental condition was imaged at least in triplicate. Image analysis applying deconvolution algorithms was performed on the central section of the image including consistently 13 acquisition layers.

### Scanning electron microscopy (SEM)

SEM images were obtained using a FEI Verios 460 field emission scanning electron microscope at an accelerating voltage of 5 kV and a

decelerating voltage of 4 kV. The landing voltage was 1 kV. The sample for SEM imaging was prepared by dispersing the powder sample on a carbon tape and keeping it under vacuum for 1 h before the imaging. The scale bars of the obtained micrographs were post-processed for better visualization (Fig. 1a).

## Fluorescence-activated cell sorting (FACS)

Immediately before sorting, samples were filtered with a 35 mm nylon mesh using BD tubes 12 × 75 mm (BD, Germany) and analyzed using the cell sorter BD FACS Melody (BD, Germany) equipped with BD FAC-SChorus software (BD) as previously described[31,36]. By using FACS, we aimed to select the bacteria population that binds the MSNs. Briefly, the background noise of the machine and of PBS was detected using the parameters forward scatter (FSC) and side scatter (SSC). Bacteria were then displayed using the same settings in a scatter plot using the forward scatter (FSC) and side scatter (SSC) and pre-gated. Singlets discrimination was performed. MSNs resuspend in PBS and a sample incubated in the same conditions without MSNs but stained with DAPI was used to set the gate for rhodamine positive signal. Rhodamine-DAPI double positive signal corresponding to the MSNs with bacteria attached were then sorted into tubes. Then, 500,000 events were sorted for each sample. Analysis of the samples showed a purity of >99%. FACS data were further analyzed with the R package flowCore[59].

## DNA extraction, 16S rRNA gene-targeted amplicon sequencing and sequence pre-processing

DNA extraction was performed for both the total microbial community (fecal samples incubated 1 h and 0 h with MSNs) and FACS-sorted fraction (fecal samples incubated 1 h with MSNs and sorted by FACS) using the QIAamp DNA mini kit (Qiagen, Germany) following the protocols for bacteria according to the manufacturer's instructions with an additional lysozyme step (Sigma-Aldrich, Austria)[36]. Sequencing was performed at the Joint Microbiome Facility of the Medical University of Vienna and the University of Vienna (project ID JMF-2009-4) and at Microsynth (Austria). 16S rRNA genes were amplified by two-step PCR barcoding approach as previously described[60,61]. The V3-V4 regions of the 16S rRNA genes were amplified with the primers 341F (5′-CCT ACG GGN GGC WGC AG-3′) and 785R (5′-GAC TAC HVG GGT ATC TAA TCC-3′) containing 16 bp head adapters (H1: 5′-GCTATGCGCGAGCTGC-3′, H2: 5′-TAGCGCACACCTGGTA-3′) and used in the first PCR step. Samples were then purified and normalized using a SequalPrep™ Normalization Plate Kit (Invitrogen). Afterward, a second barcoding PCR step was performed with unique dual barcodes (UDBs; Pjevac et al.[60]). Samples were again purified and normalized using a SequalPrep™ Normalization Plate Kit and pooled and concentrated on columns (innuPREP PCRpure Kit, Analytik Jena). Next, sequence libraries were prepared with the Illumina TruSeq DNA Nano Kit and sequenced in paired-end mode (2 × 300 nt; v3 chemistry) on an Illumina MiSeq. After sequencing, amplicon pools were extracted from the raw sequencing data using the FASTQ workflow in BaseSpace (Illumina) with default parameters, and then sequences were demultiplexed with the python package demultiplex by permitting one mismatch each for barcodes, linkers, and primers[60,61]. 16S rRNA gene sequence data were processed into amplicon sequence variants (ASVs) using the Divisive Amplicon Denoising Algorithm (DADA2)[62] applying the recommended workflow[63]. FASTQ reads 1 and 2 were trimmed at 250 nt and 200 nt with allowed expected errors of 4 and 6, respectively. ASV sequences were subsequently classified using DADA2 and SILVA database SSU Ref NR 99 release 138 (https://doi.org/10.5281/zenodo.3986799).

Sequences from contaminants were removed using the R package decontam v1.6.0 using the default threshold value of 0.1 for the prevalence-based statistical test[64].

To avoid biases in downstream analysis related to uneven library depth, sequencing libraries were subsampled to a number of reads

smaller than the smallest library (1000 reads); 1000 reads were sufficient to maintain a high coverage per library (mean: 98%). 16S rRNA gene sequence data has been deposited in the NCBI Short Read Archive under PRJNA718139 (https://www.ncbi.nlm.nih.gov/sra/?term=PRJNA718139).

## Anaerobic incubations

Eleven fresh stool samples were introduced and processed in an anaerobic tent (85% $N_2$, 10% $CO_2$, 5% $H_2$) immediately after collection. Inulin (2 mg/ml) (Sigma-Aldrich, Austria), fructose (2 mg/ml) (Carl Roth, Germany), FOS (2 mg/ml) (Sigma-Aldrich, Austria), XOS (2 mg/ml) (Carl Roth, Germany) and no amendment control (nothing added) were used as amendments for the incubations. In this work, we did not compare different concentrations since 2 mg/ml is in line with nutritional recommendations[5,34]. 2X PBS was added to the fecal sample and the mixture was vigorously vortexed for 2–3 min until homogenized. Afterward, the samples were filtered using a 40 μm size filter (Corning, Germany) to remove large particles, then transferred to a new auto-claved tube and diluted 1:10 with 2X PBS. Subsequently, 2 ml of sub-strates in $D_2O$ (heavy water) were transferred into autoclaved Hungate tubes, incubated with 2 ml of fecal sample and 5 mM non-canonical amino acid L-azidohomoalanine (AHA) (Baseclick, Germany). The samples were incubated at a final volume of 4 ml in the presence of AHA (final concentration: 50 μM) and $D_2O$ (final concentration: 50%). Samples were incubated in an anaerobic tent at 37 °C for 6 h. At the end of the incubation, the biomass was washed with PBS to remove traces of $D_2O$ and AHA, and subsamples of the biomass (1 ml) were washed twice in PBS, fixed in ethanol: PBS (1:1) and stored at −20 °C. Further sample aliquots of 1 ml each were collected for nucleic acid extraction and metabolite analysis, and stored at −80 °C until further use. Samples were further stored in 20% glycerol/PBS in crimp-sealed vials with rubber stoppers and stored at −80 °C for further RACS experiments.

## Bioorthogonal non-canonical amino acid tagging (BONCAT), FACS, DNA extraction and 16S rRNA gene-targeted amplicon sequencing

Cu(I)-catalyzed click labeling of chemically fixed microbial cells was performed in solution according to Hatzenpichler et al.[35] right before sorting the cells with FACS[36]. The azide-alkyne click reaction was achieved via a Cu(I)-catalyzed reaction where a terminal alkyne coupled to the Cy5 alkyne fluorescence dye (Jena Bioscience, Germany) was linked to the azide group of AHA yielding a triazole[35].

A representation of BONCAT-inulin-positive cells is shown in Supplementary Fig. 5b. For flow cytometry sorting, bacteria were labeled in Cy5 dye and sorted as previously described[36] (Supplementary Fig. 5e–h).

Absolute cell count was also performed with the cell sorter FACS Melody (BD, Germany), and BONCAT-positive cells were counted in triplicate for each sample (Supplementary Fig. 5d). Absolute counting beads (CountBright™, Invitrogen, ThermoFisher Scientific, Austria) were used for cell counts according to the manufacturer's instructions.

Bacterial DNA from both the total community (fecal samples incubated 0 h and 6 h with inulin, XOS FOS and fructose) and FACS-sorted cells (fecal samples incubated 6 h with inulin, XOS, FOS and fructose and sorted by FACS) were extracted with a QIAamp DNA Mini Kit (Qiagen, Germany) following the manufacturer's instruction with an additional lysozyme step (Sigma-Aldrich, Austria). PCR was performed with a two-step barcoding approach and sequence data were pre-processed as described above[60]. 16S rRNA gene sequence data has been deposited in the NCBI Short Read Archive under PRJNA718139.

## Raman microspectroscopy

Single-cell Raman spectra were measured from all fecal samples amended with inulin and XOS and incubated for 6 h. To obtain Raman spectra of individual cells, 1.5 μl of each previously fixed sample

(ethanol:PBS, 1:1) was directly spotted on an aluminum-coated slide (Al136; EMF Corporation, USA), washed by dipping into ice-cold Milli-Q (MQ) water (Millipore, Austria) to remove traces of buffer components, and air-dried. Single microbial cell spectra were acquired using a confocal Raman microspectroscope (LabRAM HR800, Horiba Scientific, France) equipped with a 532 nm neodymium-yttrium aluminum garnet (Nd:YAG) laser and 300 grooves/mm diffraction grating. Spectra were acquired with the software LabSpec 6 in the range of 400–3200 cm$^{-1}$ for 30 s and for each sample, 35–40 individual cells were measured. For quantification of the degree of D substitution in C-H bonds (%CD), the peaks assigned to the C-D (2040–2300 cm$^{-1}$) and C-H (2800–3100 cm$^{-1}$) bonds were calculated using integration of the specified region with the single-cell analysis and testing tools for Raman microspectroscopy (Scattr) (http://shiny.csb.univie.ac.at:3838/scattr/)[37].

### Raman-activated cell sorting (RACS) and targeted isolation

RACS was performed following the design and working principle of the system as described by Lee et al.[38,65] Samples were incubated in D$_2$O-containing medium for 6 h in the presence of inulin and stored in 20% glycerol (balanced with PBS) at −80 °C. For sorting, the cells were thawed, pelleted by centrifugation (7 min, 7000–9000×g), washed twice with 0.3 M glycerol (balanced with MQ water; to minimize the osmotic stress on the cells), and then resuspended in 500 µl of 0.3 M glycerol. Cells of interest were identified and sorted using a platform that combines the Raman microspectroscope (532 nm at 90 mW), optical tweezers (1064 nm Nd:YAG laser at 500 mW), and a polydimethylsiloxane (PDMS) microfluidic device[38,65].

This microfluidic system has two outlets (each for a waste and a collection) and the sample stream is engineered to exit through the waste outlet by default. Individual cells are randomly captured within the sample stream using the optical tweezers and their Raman spectra are measured. Cells identified as D-labeled are translocated to a sample-free region (within the microfluidic device) and then released from the optical tweezers so that the flow carries them to the collection outlet for the collection, whereas cells identified as unlabeled are immediately released (without the translocation) to exit through the waste outlet and discarded. This process is fully automated using an in-house software written in MATLAB (version 4.2) and to this end the cell index $P_C$ (which detects the capture of cells in the optical tweezers; $P_C = I_{1620-1670}/I_{fluid,1620-1670}$, where $I$ refers integrated intensity within the specific spectral window; $P_C > 1$ upon the cell capture) and the labeling index $P_L$ (which differentiates between D-labeled and unlabeled cells; $P_L = I_{2040-2300}/I_{1850-1900}$) are employed. The threshold value for $P_L = 5.7$ was chosen on the basis of mean + 2 sd of the control sample incubated with inulin in a non-D$_2$O-containing medium (i.e., water control)[38,65]. After approximately 60–90 min, sorting was stopped and 50–60 µl liquid (containing the sorting cells) from the collection tube was collected in an Eppendorf tube, immediately introduced into an anaerobic tent and plated on rich media supplemented with 5 mg/ml glucose (YCFA-G, DSMZ 1611). Sorted samples were incubated at 37 °C until colony appearance on the plate. Single colonies were streaked on a new agar plate with YCFA-inulin (YCFA-IN) and incubated at 37 °C until new colonies were formed. For all sort experiments, the buffer 0.3 M glycerol/MQ water utilized was checked for purity and thus plated on YCFA-G overnight. No colony growth was detected.

### Colony PCR and Sanger sequencing

The 16S rRNA genes from resulting colonies YCFA-IN or YCFA-XOS selective agar plates were amplified by colony PCR using the primers 616V (5'- AGA GTT TGA TYM TGG CTC AG-3') and 1492R (5'-GGT TAC CTT GTT ACG ACT T-3'). Colonies from YCFA-G that were plated on YCFA-IN or XOS were also sequenced as quality control to ensure that the isolates were pure cultures. A single colony was picked from the

plate with the tip of the loop and dissolved in the PCR reaction mix. The PCR reaction mix (final concentrations; Green 1X Dream Taq Buffer, dNTPS 0.2 mM, BSA 0.2 mg/ml, Taq polymerase 0.05 U/µl, primers 1 µM) was prepared in a final volume of 50 µl per reaction. The amplification cycles were as follows: initial denaturation at 95 °C for 3 min, followed by 30 cycles at 95 °C for 30 s, 56 °C for 30 s, 72 °C for 1.5 min, and a final elongation at 72 °C for 10 min.

PCR products were visualized on 1.5% agarose gel electrophoresis and subsequently purified using a QIAquick PCR Purification Kit (Qiagen, Germany) according to the manufacturer's instructions. Concentrations were measured by Nanodrop, and samples were sent for Sanger sequencing at Microsynth AG (Vienna, Austria). Sorted bacterial strains were identified by analyzing the obtained sequencing results in RDP seqmatch (https://rdp.cme.msu.edu). 16S rRNA gene sequence data of the strains isolated has been deposited in the GenBank with accession numbers OK067598-OK067669.

### Cultivation in supplemented media: growth curves

All strains were grown in both solid and liquid YCFA medium until the early stationary phase and growth data were recorded (Supplementary Table 3). To perform growth curves each individual strain identified by Sanger sequencing was grown in YCFA-G as starting broth at 37 °C in an anaerobic tent. When cultures reached their maximum OD$_{600}$, they were washed and diluted into the YCFA (DSMZ 1611) broth supplemented with inulin or XOS (YCFA-IN, YCFA-XOS) and in YCFA media without any amendment (YCFA-NA) which was used as a negative control, and dispensed into wells of a sterile 96-well flat-bottomed microtiter plate (Costar 3595, Corning, NY, USA). *E. lenta* and *G. urolithinfaciens* were also grown in modified YCFA DSMZ 1611 in the presence of 10 mg/ml arginine according to Noecker et al.[55] The starting OD$_{600}$ was ~0.05. The prepared plates, which also included negative control media (YCFA-IN, XOS and NA without the addition of bacteria), were incubated for 120 h at 37 °C in a microplate reader (MultiskanTM GO Microplate Spectrophotometer, ThermoFisher Scientific) and placed in an anaerobic tent using the SkanIt software (ThermoFisher Scientific). The OD$_{600}$ was recorded automatically every 30 min for a total of 120 h with a shaking of 5 s between readings. Each condition was tested in at least three technical replicates.

Afterward, growth curves were calculated using R statistical software (https://www.r-project.org/).

In addition, cell supernatants were preserved at −80 °C for further thin layer chromatography (TLC) and fructan assay analysis.

### Degree of polymerization evaluation by thin layer chromatography (TLC)

Supernatant from stool samples incubated for 6 h with inulin and RACS cultures were applied to 20 × 10 cm HPTLC silica gel 60 plates (Merck KGaA, Germany) using a Linomat 5 (Camag AG, Switzerland). Then, 4 µl of sample (supernatant) or 2 µl of standard (fructose, glucose, sucrose, 1-kestose, and 1,1, kestotetraose at 2 mg/ml each) were applied per lane. The eluent was composed of 1-butanol:1-propanol:ethanol:water−2:3:3:2; and the plate was developed three times. After developing, the carbohydrates were derivatized by spraying the plate with aniline-diphenylamine reagent (diphenylamine 2%$_{w/v}$, aniline 2%$_{w/v}$ and phosphoric acid 15%$_{v/v}$ dissolved in acetone) and heating at 100 °C for 5 min[66,67]. The TLC plates were scanned and analyzed by densitometry with Gel Analyzer 19.1.1 (www.gelanalyzer.com). By using FOS and inulin as standards, the maximum density was used as the retention factor (Rf) value utilized to create the degree of polymerization scale seen in Supplementary Fig. 10. The presence of saccharide was deemed positive when the height of the density peak was 4x that of the baseline noise. *Phocaeicola vulgatus* (DSM1447) and *Bacteroides uniformis* (DSM6597), which encode inulinases, were used as positive controls[68].

## Fructan assay

Inulinase activity was performed by measuring the remaining inulin after incubation by using the Fructan Assay kit (K-FRUC, Megazyme, Ireland) according to the manufacturer's instructions. Briefly, to 200 μl of culture supernatant after 6 h treatment with inulin was added 200 μl of diluted enzyme solution (sucrase, β-amylase, pullulanase and maltase) and incubated at 30 °C for 30 min with the aim to hydrolyze sucrose, starch and other polysaccharides. Afterward, 200 μl of alkaline borohydride solution was added to the tube. The mixture was incubated at 40 °C for 30 min to effect complete reduction to sugar alcohols. Subsequently, 500 μl of 200 mM acetic acid was added to the tube with vigorous stirring on a vortex mixer. Then, 200 μl aliquots of the previous solution were transferred in triplicate to new tubes and 100 μl of fructanase solution (containing a mixture of endo and exo-inulinases) were added to the samples (and 100 μl of 100 mM sodium acetate buffer as blanks). The tubes were incubated at 40 °C for 30 min to reach complete hydrolysis of inulin to D-fructose and D-glucose. Finally, 500 μl of 4-Hydroxybenzhydrazide (PAHBAH) reagent was added to all tubes [(samples, blanks, inulin control (2 and 5 mg/ml) and the fructose standard)] and incubated in a boiling water bath for 6 min. Afterward, the tubes were removed from the boiling water bath and immediately placed in cold water (18–20 °C) for 5 min. The absorbance was read at 410 nm against the reagent blank and each sample was measured in triplicates.

## Fluorescence in situ hybridization (FISH)

Fecal samples were fixed in ethanol and stored in ethanol-PBS (1:1) at −20 °C.

FISH was performed using a standard protocol[69] with the addition of two permeabilization steps[70]. Briefly, 5 μl of the fixed sample was spotted on microscopy slides (Marienfeld, Germany) and dried in a humidified chamber at 46 °C for 5 min. The samples were then submitted to a dehydration step in an ethanol series (50-80-96%) for 3 min each. Samples were permeabilized in a humid chamber with lysozyme (10 mg/ml dissolved in 0.05 M EDTA pH:8.0 and 0.1 M Tris-HCl pH:7.4) and incubated for 60 min at 37 °C. Subsequently, samples were treated with 60 U/ml achromopeptidase (Sigma-Aldrich, Austria) dissolved in 0.01 M NaCl and 0.01 M Tris-HCl, pH: 8.0 and incubated for 30 min at 37 °C. After permeabilization samples were washed in MQ water[70]. Afterward, 10 μl of hybridization buffer containing 10 or 5% formamide was applied with subsequent addition of 5 μM of specific probes [E.len194: 5′-CCTTGCCGTCTGGGCTTT-3 (*Eggherthella lenta*)[71], COR653: 5′-CCCTCCCMTACCGGACCC-3 (*Collinsella* and *Coriobacterium*)[72], EUBmix338-I: 5′GCTGCCTCCCGTAG-GAGT-3′ (most bacteria), EUBmix338-II:5′GCAGCCACCCGTAGGTGT-3′ (*Planctomycetales*), and EUBmix338-III: 5′-GCTGCCACCCGT AGGTGT-3′ (*Verrucomicrobiales*)][73,74]. Specifically, a 10% formamide hybridization buffer was used for E.len194[71] and 5% for COR653[72]. After 2 h of hybridization in a humidified chamber at 46 °C, the slides were washed in the respective wash buffers and counterstained with 4′, 6-diamidino-2-phenylindole (DAPI). The NONEUB probe (5′ACTCCTACGGGAGGCAGC-3′) was used as a negative control for Cy3 and FLUOS-dyes[75] and *Eggerthella lenta* pure culture isolated with RACS was used as a positive control (Supplementary Figs. 11 and 12). FISH images were taken with a confocal scanning laser microscope (Leica TCS SP8X, Mannheim, Germany) equipped with an Ar-laser (495 nm) for excitation of the FLUOS-dyes and He-Ne-lasers (550 nm) for excitation of Cy3. Pictures were acquired with 63X or 93X objectives, pinhole size of 1 μm, resolution between 1024 × 1024 and 2396 × 2396 pixels and zoom factor of 1. Confocal microscopy images were collected and the biovolume fraction of *Collinsella* and *Coriobacterium* was calculated using the software daime[76].

## Liquid-FISH combined with Raman microspectroscopy

Fecal samples fixed in ethanol: PBS (1:1) were used for liquid FISH analysis combined with Raman spectroscopy analysis. Liquid FISH

was performed using a standard protocol[77] and with the same probes and procedures as described above. Briefly, liquid in situ hybridization was performed resuspending the cell pellets in 10 μl hybridization buffer (HB). Then, 1 μl of the respective probe (COR653 or E.len194) was added to the mixture and incubated at 46 °C for 2 h. Afterward, samples were washed by centrifugation at 40 °C (20,000×*g* for 15 min), resuspended in 50 μl of pre-heated (48 °C) washing buffer (WB) and incubated at 46 °C for 15 min. Subsequently, the samples were centrifuged again at 40 °C (20,000×*g* for 10 min) and resuspended in ice-cold 1X PBS. For Raman measurements, 1.5 μl of sample was applied on an aluminum slide and dried at 46 °C for 5 min as described above. The slide was placed under a 100X air objective and Raman spectra were recorded for randomly selected fluorescence-positive cells and were measured with an epifluorescence microscope combined with a Raman microscope (LabRAM HR800, Horiba Scientific, France). Then, 35–40 measurements were collected for each sample.

## Isolation of genomic DNA and genomic sequencing

Isolation of genomic DNA of *Eggerthella lenta* 6.2 and *Gordonibacter urolithinfaciens* AL-11 was performed using the genomic DNA purification kit according to the manufacturer's instructions (Promega, Germany). Briefly, 1 ml of culture was centrifuged at 13,000–16,000×*g* for 2 min to pellet the cells. Cells were resuspended in 50 mM EDTA and 10 mg/ml lysozyme and incubated at 37 °C for 30–60 min. Afterward, cells were centrifuged for 2 min at 13,000–16,000×*g*, the supernatant was removed and cells were resuspended in lysis buffer and incubated at 80 °C for 5 min to further lyse the cells. Then, 3 μl of RNase solution was added to the cell lysate and incubated at 37 °C for 15–60 min. Next, 200 μl of protein precipitation solution was added to the RNase-treated cell lysate and incubated on ice for 5 min. Cells were centrifuged at 13,000–16,000×*g* for 3 min and isopropanol was added to the cell supernatant. Tubes were mixed by inversion until the thread-like strands of DNA were forming a visible mass. Subsequently, samples were centrifuged at 13,000–16,000×*g* for 2 min and 70% ethanol was added to the samples. Tubes were gently inverted several times to wash the DNA pellet. The DNA was centrifuged at 13,000–16,000×*g* for 2 min to remove the ethanol. Pellets were air-dried for 10–15 min at room temperature. Finally, 50 μl of DNA was eluted and DNA was rehydrated by incubating at 65 °C for 1 h and afterward stored at −20 °C.

Sequencing libraries were prepared with the NEBNext® Ultra™ II FS DNA Library Prep Kit for Illumina® (NEB) and sequenced on the Illumina MiSeq platform (2 × 300 bp v3 chemistry) following the manufacturer's instructions. Sequencing was performed at the Joint Microbiome Facility of the Medical University of Vienna and the University of Vienna (project ID JMF-19DM-2).

Sequencing data was extracted using the FASTQ workflow in BaseSpace (Illumina) with default parameters. Thereafter, adapter contaminations were removed and the data was quality filtered at a phred score of 15 using the bbduk function of BBMap (v36.20, https://github.com/BioInfoTools/BBMap). The quality trimmed sequence data was used as input for genome assembly with SPAdes[78], where kmers were iterated in steps of 10 between 11 and 121. Assembly quality was inspected with CheckM[79].

Annotation was performed with the RASTtk pipeline/annotation server using the default settings (https://rast.nmpdr.org/rast.cgi) and with EggNOG[80]. The whole genome project has been deposited at NCBI under the accession JAIPUQ000000000 for *Eggerthella lenta* 6.2 and JAIPUP000000000 for *Gordonibacter urolithinfaciens* AL-11 (PRJNA718139). Genomes were visualized with GCView Server[81]. Average nucleotide identity (ANI)[82] was calculated for *Eggerthella lenta* 6.2 and *Gordonibacter urolithinfaciens* AL-11 genomes in comparison to publicly available reference genomes from the respective genera (Supplementary Table 4).

## Phylogenetic tree reconstruction

Downloaded 16S rRNA sequences and RACS-derived 16S rRNA sequences were clustered at 100% identity using Usearch[83] into representative centroids. Representative centroids were first length filtered (>1200 bases) and then aligned using Mafft-linsi[84]. Trimal[85] was used to trim the ends of the alignment (-nogaps -terminalonly) and to remove poorly aligned internal sites (-gt 0.95). IQtree[86] was used for the phylogenetic reconstruction of a reference tree under the GTR model with branch support evaluated with 1000 ultrafast bootstraps[87]. Blastn[88] was used to identify perfect matches between ASVs and sequences used for phylogenetic analysis. ASVs and short (<1200 nt) RACS/representative sequences were added to the alignment of trimmed sequences (>1200 nt) that were used for phylogenetic reconstruction using mafft (--addfragments, --keeplength) and placed into the reference tree using RAxML EPA[89]. An additional chimera check was performed using the online tool DECIPHER[90]. Finally, the phylogenetic trees were visualized and plotted using the R package ggtree[91] and iTOL[92].

## High-performance anion-exchange chromatography (HPAEC)

XOS strains isolated with RACS were grown in XOS-supplemented media for 24 h. Sample supernatants were collected and analyzed by HPAEC (Dionex ICS 3000, ThermoFisher, Germany). The sugars were analyzed on a Dionex CarboPac PA20 (3 × 150 mm) column with a Dionex CarboPac PA20 (3 × 30 mm) guard column (solvent: 100 mM KOH, 35 min at 0.3 ml/min).

The standard carbohydrate waveform was used for detection, and the analyses were carried out in duplicate. The standards used for the construction of the calibration curve were: D-xylose (Sigma-Aldrich, St. Louis, MO, USA), xylobiose and xylotriose (Megazyme Ltd, Bray, Ireland). The XOS (Carl Roth) used for the experiments was composed of xylose (% w/w, 0.1%), xylobiose (38.9%), xylotriose (32.7%), xylotetraose (18.6%), xylopentaose (9.8%) as measured with HPAEC.

## Statistical analysis

Statistical analysis was performed using R statistical software (https://www.r-project.org/) and GraphPad Prism version 7 (www.graphpad.com). Statistical analysis to compare sample groups was performed using ANOVA, Student's $t$-test, and the R package DESeq2 (v.1.30.1) that estimates variance-mean dependence in count data from high-throughput sequencing assays and test for differential expression based on a model using the negative binomial distribution[93,94]. The statistical significance of factors affecting microbiota composition and differences between time points was evaluated using non-parametric permutational multivariate analysis of variance (PerMANOVA), significant clustering of groups was evaluated with analysis of similarities (ANOSIM), ordination was performed using principal component analysis (PCoA) and non-metric multidimensional scaling (NMDS) in the vegan package (v.2.5-7) (https://cran.r-project.org/web/packages/vegan/vegan.pdf). Jackknife diversity index and Bray-Curtis distances were also calculated in the vegan package. Variables were expressed as mean (sd, standard deviation) and a probability value ($p$-value) less than 0.05 was considered statistically significant and adjusted $p$-value with False Discovery Rate method (FDR) were used for multiple comparison.

## Data availability

16S rRNA gene amplicon sequence data from FACS and RACS experiments have been deposited in the NCBI Short Read Archive under PRJNA718139. *Eggerthella lenta* 6.2 and *Gordonibacter urolithinfaciens* AL-11 genomes were deposited at NCBI under accession JAIPUQ000000000 and JAIPUP000000000 respectively and under PRJNA718139. 16S rRNA gene sequences data of the strains isolated with RACS after inulin supplementation have been deposited in GenBank with accession numbers OK067598-OK067669 and RACS strains sequences isolated after XOS supplementation have been deposited in GenBank with accession numbers OP183499-OP183547 (https://www.ncbi.nlm.nih.gov/genbank/). Source data are provided with this paper.

## Code availability

MATLAB script and graphical user interface (version 4.2) code for the operation of the RACS platform are provided in Supplementary Codes 1 and 2.

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

## Acknowledgements

This work was financially supported by the European Research Council (Starting Grant: FunKeyGut 741623, D.B.), the Austrian Science Fund (P27831-B28, D.B.; Cluster of Excellence COE7, D.B., C.K., P.P., A.Ric., M.Wag.), the United States Department of Energy (DE- SC0019012, D.B.), and the Novo Nordisk Foundation (NNF21OC0066551, M.V.B and N.J.). C.V.B. and F.K. acknowledge the funding support of the University of Vienna (Austria). M.V.B and N.J. thank the Helmholtz Centre for Environmental Research (Germany) for the ProMetheus platform for proteomics and metabolomics. We thank Jasmin Schwarz, and Gudrun Kohl from the Joint Microbiome Facility (University of Vienna, and Medical

University of Vienna, Austria) for support in the genomic and 16S rRNA gene amplicon library preparation and the Core Facility MULTIMODAL IMAGING member of Vienna Life Science Instruments (VLSI) for supporting the recording of the confocal microscopy images of MSNs.

## Author contributions

A.R. and D.B. conceived and designed the study and the experiments. A.R. performed all the experiments and the data analysis. A.R. and D.B. wrote the paper. J.M.C.R. and H.V. performed TLC and supported TLC data analysis and evaluation. H.R., D.I., N.H., J.W. and C.K. performed the experiments with the prebiotic XOS. S.L.S. and C.W.H. performed bioinformatic analysis and phylogeny. C.V.B. and F.K. prepared and provided mesoporous silica nanoparticles. G.D.F. performed structure illumination images (SIM). B.H. and P.P. performed genomic analysis and supported other bioinformatic analyses. M.P., K.S.L., M.Wag. and R.S. supported the Raman microspectroscopy experiments. A.Sp. supported FACS analysis. A.Sc., M.Wag., M.Wat., N.J., M.V.B. and A.Ric. supported in data contribution and evaluation. G.N., S.K., and A.K. supported molecular biology experiments. All authors have approved the final version of the paper.

## Competing interests

The authors declare no competing interests.

## Additional information

[1]Centre for Microbiology and Environmental Systems Science, Department of Microbiology and Ecosystem Science, Division of Microbial Ecology, University of Vienna, Vienna, Austria. [2]Doctoral School in Microbiology and Environmental Science, University of Vienna, Vienna, Austria. [3]Department of Pharmaceutical Technology and Biopharmaceutics, University of Vienna, Vienna, Austria. [4]Department of Functional Materials and Catalysis, Faculty of Chemistry, University of Vienna, Vienna, Austria. [5]Joint Microbiome Facility of the Medical University of Vienna and the University of Vienna, Vienna, Austria. [6]Department of Laboratory Medicine, Medical University of Vienna, Vienna, Austria. [7]Core Facility Flow Cytometry and Surgical Research Laboratories, Medical University of Vienna, Vienna, Austria. [8]Department of Food Chemistry and Toxicology, Faculty of Chemistry, University of Vienna, Vienna, Austria. [9]Helmholtz Centre for Environmental Research, Department of Molecular Systems Biology, Leipzig, Germany. [10]Centre for Microbiology and Environmental Systems Science, Department of Microbiology and Ecosystem Science, Division of Terrestrial Ecosystem Research, University of Vienna, Vienna, Austria. [11]Institute for Environmental Engineering, Department of Civil, Environmental and Geomatic Engineering, ETH Zurich, Zurich, Switzerland. [12]Center for Microbial Communities, Department of Chemistry and Bioscience, Aalborg University, Aalborg, Denmark. [13]Present address: Chair of Nutrition and Immunology, School of Life Sciences, Technical University of Munich, Freising-Weihenstephan, Germany. ✉e-mail: david.berry@univie.ac.at

