## [Peer Review File · Nature Communications]

REVIEWERS' COMMENTS

Reviewer #1 (Remarks to the Author):

The modifications to the manuscript have appropriately addressed reviewer concerns. It will be exciting to see how these methods can be used to advance gut microbiome research.

Reviewer #2 (Remarks to the Author):

I am happy that the authors have addressed all of my outstanding comments in the revised MS and have removed reference to *E. lenta* and *G. urolithinifaciens* as inulin users when this conclusion was not supported by the data presented.

I have only a couple of minor comments to address on the revised MS:

1. Line 306 – ‘As for inulin, XOS-supplemented media boosted the growth of all the isolates’ this isn’t true for inulin (line 249) – would be better to read as ‘Unlike for inulin, XOS-supplemented media boosted the growth of all the isolates’

2. Line 350 – ‘At best of our knowledge, it has not yet known how inulin is imported in the cells, but it has only been proposed in *Bacteroides* spp. a mechanism of import across the outer membrane without surface pre-digestion.’ Not really clear what this means. Polysaccharide import in *Bacteroides* spp (and likely all *Bacteroidetes* as all studied so far have SusCD homologous in PULs) is well studied – see e.g. White et al 2023 (PMID: 37286596) and refs therein. While inulin is short enough on average to be imported without the need to pre-digestion (see e.g. Ref 23) it will still require a SusCD TBDT to transport across the OM. In other Gram positive members of the microbiota polysaccharide transport is less well studied but ABC transporters have been implicated in several

studies with Firmicutes (see e.g. Leth et al 2018 PMID: 29610517) and Bifidobacteria (e.g. Fushinobu and Abou Hachem 2021 PMID: 29610517).

REVIEWERS' COMMENTS

Reviewer #1 (Remarks to the Author):

The modifications to the manuscript have appropriately addressed reviewer concerns. It will be exciting to see how these methods can be used to advance gut microbiome research.

Reviewer #2 (Remarks to the Author):

I am happy that the authors have addressed all of my outstanding comments in the revised MS and have removed reference to *E. lenta* and *G. urolithinifaciens* as inulin users when this conclusion was not supported by the data presented.

I have only a couple of minor comments to address on the revised MS:

1. Line 306 – ‘As for inulin, XOS-supplemented media boosted the growth of all the isolates’ this isn’t true for inulin (line 249) – would be better to read as ‘Unlike for inulin, XOS-supplemented media boosted the growth of all the isolates’

This has been corrected as suggested by the reviewer.

2. Line 350 – ‘At best of our knowledge, it has not yet known how inulin is imported in the cells, but it has only been proposed in *Bacteroides* spp. a mechanism of import across the outer membrane without surface pre-digestion.’ Not really clear what this means. Polysaccharide import in *Bacteroides* spp (and likely all Bacteroidetes as all studied so far have SusCD homologous in PULs) is well studied – see e.g. White et al 2023 (PMID: 37286596) and refs therein. While inulin is short enough on average to be imported without the need to pre-digestion (see e.g. Ref 23) it will still require a SusCD TBDT to transport across the OM. In other Gram positive members of the microbiota polysaccharide transport is less well studied but ABC transporters have been implicated in several studies with Firmicutes (see e.g. Leth et al 2018 PMID: 29610517) and Bifidobacteria (e.g. Fushinobu and Abou Hachem 2021 PMID: 29610517).

We agree with the reviewer. We have shortened this sentence to: „ Polysaccharides can be either hydrolyzed by secreted enzymes or captured by cell surface proteins with carbohydrate-binding modules prior to import²⁶⁻²⁹.”